# Catalytic Ozonation of Organics in Reverse Osmosis Concentrate with Catalysts Based on Activated Carbon

**DOI:** 10.3390/molecules24234365

**Published:** 2019-11-29

**Authors:** Xieyang Xu, Zhilin Xia, Laisheng Li, Qi Huang, Can He, Jianbing Wang

**Affiliations:** 1School of Chemical and Environmental Engineering, China University of Mining and Technology (Beijing), Beijing 100083, China; xuxy_env@163.com (X.X.); zlxia_st@rcees.ac.cn (Z.X.); hecan086@163.com (C.H.); 2School of Chemistry & Environment, South China Normal University, Guangzhou 510006, China; llsh@scnu.edu.cn; 3Beijing Shuimengyuan Water Science and Technology Co. LTD, Beijing 100019, China; 90570618@163.com

**Keywords:** catalytic ozonation, reverse osmosis concentrate, activated carbon, cerium

## Abstract

Acid-washed activated carbon (AC-A), nitric acid modified activated carbon (AC-NO_2_), aminated activated carbon (AC-NH_2_) and cerium-loaded activated carbon (Ce/AC) were prepared and characterized by BET procedure, Boehm titration and SEM. Their performances were investigated for the ozonation of p-chlorobenzoic acid (p-CBA) in its solution and organic compounds in reverse osmosis concentrate (ROC). Nitration and amination had little effect on the surface area of catalyst, but increased the concentration of surface acid and basic functional group respectively. After loading Ce, the surface area of the catalyst decreased, and amount of Ce particles were agglomerated on the surface of activated carbon. All the four catalysts can improve the removal rate of the organics in water. Among the four catalysts, Ce/AC shows the highest catalytic activity. The removal rates of p-CBA, TOC and three target pollutants (e.g., tetracycline, metoprolol, atrazine) are 99.6%, 70.38%, 97.76%, 96.21% and 96.03%, respectively. Hydroxyl radical (·OH) was proved to be the core of catalytic reaction mechanism for Ce/AC, with the contribution rate to p-CBA removal of 91.4%. The surface groups and the Ce loaded on AC were the initiator for the rapid generation of ·OH. Electron transfer between electron-rich structures and cerium oxide might be a synergistic effect that can increase catalytic activity of Ce loaded on AC. Catalytic ozonation with Ce/AC is a promising ROC treatment technology due to its efficiency and possibilities for improvement.

## 1. Introduction

Due to its availability and reliability, reverse osmosis (RO) process has been widely applied in municipal wastewater treatment plants (WWTP) for water reclamation. When 75–85% of feed water was reclaimed, the reverse osmosis concentrate (ROC) was generated concurrently, containing salts and organics in high concentration [1]. In addition, various emerging pollutants (e.g., pesticides, personal care products, pharmaceutical products, endocrine disruptors) were detected in the ROC, which are reported to be refractory and toxic [2,3,4]. If ROC was directly discharged to the aquatic environment, these refractory organics will cause severe ecotoxicological risk, although they were in low concentrations (ng/L). Consequently, it is vital to explore appropriate processes for the effective abatement of organics in ROC.

Ozonation has been widely used in the treatment of ROC and degradation of refractory organics for its convenience and cleanliness [5]. Regrettably, TOC removal rate of ROC treatment by ozonation is very low [5], which is mainly due to some intermediates, such as aldehydes and carboxylic acids, that can hardly be oxidized for the selectivity of ozonation [6]. Moreover, some by-products, such as bromate, aldehydes and organic bromine, will be generated during ozonation and result in the increase of cytotoxicity and genotoxicity [7,8].

Heterogeneous catalytic ozonation has been proved to be more effective in the degradation of various target pollutants than ozonation [9,10]. In addition, this technique has been shown to inhibit the formation of bromate [11,12]. Carbon material [13,14,15], metal material [16,17] and their combinations [18,19,20] are commonly used as catalysts in this process. The catalytic activity of carbon materials depends on the basal plane and functional groups. Basal plane [21], basic groups [14] and deprotonated oxygenic acid [22,23] are all considered to be able to decompose O_3_ into ·OH. Metal oxides could absorb O_3_ with surface oxygen vacancies (OVs) and transform it into ·OH [9]. Among various metal oxides, cerium oxide has a good catalytic activity because of the redox couple (Ce^4+^/Ce^3+^) [24]. 

Although the availability of catalytic ozonation for removing refractory organics in water have been proven in previous literatures, there are few studies on the treatment of ROC by catalytic ozonation with efficient carbon material catalyst. Also, great efforts are still needed to explore the mechanism for the removal of refractory organic compounds from water by ozonation. In this study, acid-washed activated carbon (AC-A), nitric acid modified activated carbon (AC-NO_2_), aminated activated carbon (AC-NH_2_) and cerium-loaded activated carbon (Ce/AC) were prepared and their performance in the ozonation of p-CBA and organic compounds in ROC was investigated. A possible mechanism was proposed for the catalytical ozonation of organic compounds in water.

## 2. Results and Discussion

### 2.1. Characterization of the Catalysts 

Table 1 shows the BET analysis results of various catalysts. Modification by nitration posed negligible effect on the surface area of activated carbon while amination resulted in a slight increase in surface area. A previous study reported that surface area decreased significantly after nitration and was restored after amination [22]. The occupation and removal of functional groups (such as -NO_2_ and -COOH) on the entrance of micropores was considered a possible reason for the surface area change [22]. Other literatures concluded that the change of surface area after modifications depends on the type of carbon material and the reaction conditions [25]. Therefore, the modified functional groups by nitration and amination in current study were perhaps not fixed at the entrance of micropores. Loading of cerium on AC led to an obvious decrease in surface area which is about 1/6 of the surface area of AC. The mixture ratio of AC and Ce is 5:1 and will result in a dilution effect. It might be the main reason for the change of surface area. The average pore diameters (d_p_) of the four catalysts are all about 0.8 nm, indicating that the modification and the loading of Ce will not cause significant changes of pore structure.

Table 2 shows the amount of acid and basic surface group on AC-A are 194.0 and 153.8 μeq/g, respectively. After nitration, the amount of acid surface group increased to 313.4 μeq/g while the amount of basic surface group decreased to 57.7 μeq/g. After amination, they changed to 74.6 and 283.7 μeq/g, respectively. A same change trend of surface group was reported by the previous study [22]. This can be explained as nitration resulting in the generation of -NO_2_ groups by electrophilic substitution of hydrogens and carboxylic groups by oxidation of unsaturated groups [26]. The -NO_2_ groups on AC-NO_2_ were then reduced to -NH_2_ groups in the amination process, with oxygenated groups being partially reduced by NaBH_4_ [22].

Figure 1 shows the SEM and EDS analysis of AC-A and Ce/AC. Compared to AC-A, there are more particles loaded on the surface of Ce/AC after impregnation. As the EDS analysis shows, Point 2 has much higher amount of cerium and oxygen than those of Point 1, and we concluded that the particles on the surface of Ce/AC are mainly cerium oxide and they tend to form agglomerates during preparation.

### 2.2. Catalytic Ozonation of p-CBA in Water

Figure 2 provides the removal of p-CBA by various processes with four catalysts. O_3_+catalysts shows the removal of p-CBA in catalytic ozonation processes. O_3_ alone shows the influence of direct ozone oxidation and self-decomposition of ozone. O_2_+catalyst, in which ozone was replaced by oxygen, shows the effect of catalyst adsorption. O_3_+catalysts+TBA, the catalytic ozonation with the adding of TBA (20 mmol/L) as free radical scavenger, reveals the contribution of ·OH in catalytic ozonation system.

At 120 min, the removal rates of p-CBA by adsorption for these catalysts are 26.9%, 27.7%, 29.8% and 21.4%, respectively, indicating that the adsorption rate of p-CBA on AC is slow. Some references also reported a very slow adsorption kinetics of p-CBA on activated carbon [13,27]. The adsorption rate of p-CBA on the Ce/AC is slower than that on other catalysts. This is predictable because Ce/AC has the smallest surface area among these catalysts.

The removal rates of p-CBA at 120 min by catalytic ozonation with AC-A, AC-NO_2_, AC-NH_2_ and Ce/AC are 79.2%, 92.8%, 83.9% and 99.6%, respectively, which are all higher than those by ozonation, adsorption or their combination. It can be concluded that there is a synergetic effect between catalyst and O_3_. The removal rates of p-CBA with AC-NO_2_ and AC-NH_2_ were both higher than that of AC-A, which means that modification can improve the activity of AC. For AC-NO_2_, carboxylic acid groups lost protons under experimental conditions (pH = 9) and deprotonated acid functional groups can react with O_3_ and generate O_3_^−^ [23]. Then, O_3_^−^ can further generate ·OH, which are active for the oxidation of organic matter in water [6]. For AC-NH_2_, amide and pyrone are able to promote O_3_ decomposition and ·OH formation, relying on its basicity and electron donating effect [14,22]. Figure 2 also shows Ce/AC is the most active for p-CBA degradation among the four catalysts. Previous literature reported that AC_0_-Ce-O is active for the catalytic degradation of oxamic acids, oxalic acids and dyes in wastewater and the removal of organic compounds depends on surface reaction of activated carbon and hydroxyl radical oxidation which is enhanced by the synergistic effect between CeO_2_ and AC [28,29]. After adding TBA, the removal rate of p-CBA decreased for all catalysts, proving the generation of ·OH in catalytic ozonation systems. 

### 2.3. The Removal of Organics in ROC by Catalytic Ozonation

Figure 3a shows the removals of TOC for the treatment of ROC by O_3_ alone, O_3_/H_2_O_2_ and catalytic ozonation processes. About 10% of the organics in ROC had been removed by single ozonation. When AC-A, AC-NO_2_ and AC-NH_2_ were added in the ozonation system as catalysts, TOC removal rate increased to 23.7%, 20.5% and 25.3%, respectively. The increase of TOC removal rate is due to both adsorption and catalytic oxidation effect of catalyst. After adding H_2_O_2_, TOC removal rate was enhanced to 49.2%, higher than those in the catalytic ozonation processes with the AC-A, AC-NO_2_ and AC-NH_2_. It is well known that hydrogen peroxide could act as initiator of the ozone decomposition process in aqueous phase, resulting in the production of ·OH, which is very reactive for the degradation of refractory substances [6]. Ce/AC showed an excellent catalytic activity in ROC, with TOC removal rate of 70.4%, which is even higher than that of O_3_/H_2_O_2_. 

Figure 3 also shows the removal of three typical organic pollutants, tetracycline, metoprolol and atrazine from ROC by various processes. In O_3_ alone, O_3_/H_2_O_2_ and catalytic ozonation with Ce/AC, the removal rates of tetracycline at 60 min are 92.8%, 97.0% and 97.8%, respectively. The high removal rates in the three processes are mainly due to high reactivity between tetracycline ring system and O_3_ since the reaction constant between tetracycline and O_3_ is 1.9 × 10^6^ M^−1^ s^−1^ [30]. 

Metoprolol is not as easily to be degraded as tetracycline by O_3_ alone, with a removal rate of 75.2%. Both H_2_O_2_ and Ce/AC can significantly improve the removal rates of metoprolol, which are 85.6% and 96.2%, respectively. The transformation from O_3_ to ·OH might be the main reason for degradation of metoprolol in O_3_/H_2_O_2_ and catalytic ozonation because metoprolol has a much bigger reaction constant with ·OH than that with O_3_ [31]. 

Atrazine is difficult to be degraded by O_3_ alone, with a removal rate of 48.0%. Adding H_2_O_2_ can only improve the removal rate of atrazine to 76.9%. However, catalytic ozonation with Ce/AC, although a heterogeneous process, resulted in a higher removal rate of atrazine than that by O_3_/H_2_O_2_. Therefore, the reaction between ·OH and organics might not be the only physical/chemical process that was responsible for the good treat efficiency in this process. Sanchez-Polo et al. found a quickly adsorption of atrazine on activated carbon [13]. Atrazine might be adsorbed on Ce/AC during the first few minutes and then be degraded on the surface of catalysts.

### 2.4. Possible Mechanism of Catalytic Ozonation in ROC

According to the above discussion, all the four catalysts can improve the removal rates of organics in p-CBA solution and ROC. In particular, Ce/AC catalyzed ozonation shows a higher degradation rate of organic compounds than that of O_3_/H_2_O_2_ process and of other catalytic ozonation processes. Therefore, it is significant to explore the mechanism of catalytic ozonation with Ce/AC. According to the data in Figure 2, the kinetics of p-CBA removal before and after adding TBA were analyzed to investigate the contribution rate of ·OH in catalytic ozonation. The effects of adsorption and ozone oxidation are considered to be very weak in the removal of p-CBA, because p-CBA has very slow reaction kinetics with O_3_ (k_O_3__ = 0.15 M^−1^s^−1^) and slow adsorption kinetics on activated carbon but much high reaction constant with ·OH (k_OH_ = 5.2 × 10^9^ M^−1^s^−1^) [13]. The contribution rates (f, %) of ·OH were calculated according to Formula (1):f = (1 − k_TBA_/k) × 100%,(1)
k and k_TBA_ are the reaction constants of p-CBA degradation in catalytic ozonation systems before and after adding TBA. 

Table 3 shows the reaction constants (k and k_TBA_) with the present of AC-A, AC-NO_2_, AC-NH_2_ and Ce/AC and the contribution rates of ·OH in these processes. The reaction constants of p-CBA degradation without TBA are 8.20, 14.59, 8.54 and 51.52 (×10^−3^ min^−1^). After adding TBA, they decreased to 5.28, 4.24, 5.37 and 4.42 (×10^−3^ min^−1^), respectively. The contribution rates of ·OH in these four catalytic ozonation systems are 35.6%, 70.9%, 37.1% and 91.4%. It can be seen that functional groups on activated carbon can stimulate free radicals, which has been reported in the previous literature [22]. Modification by nitric acid is helpful in the production of large amounts of groups that could trigger the decomposition of O_3_ to generate ·OH, while amination may lead to the loss of these groups. After Ce was loaded, the catalyst had a high contribution rate of ·OH. It could be attributed to the functional group and the Ce particle on the activated carbon. The latter is of greater significance than the former to Ce/AC of catalytic activity. 

Ce/AC is efficient in ozonation for removal of organics mainly due to its various removal mechanisms, including adsorption, ozonation and free radical oxidation. Both adsorption and ozonation is selective. Some organics adsorb slowly on AC such as p-CBA [27] while others react slowly with O_3_, such as atrazine [13]. Therefore, hydroxyl radical oxidation, which is non-selective, plays an important role for the removal of refractory organic compounds from water by Ce/AC catalyzed ozonation. A possible mechanism for generation of ·OH in catalytic ozonation system is shown in Figure 4. In this mechanism, the generation of ·OH can be stimulated not only by surface functional groups but also Ce loaded on AC. Radical excitation by surface groups relies on its basicity and electron donating effect, which has been widely mentioned in previous studies [9,14,23]. For the Ce loaded on AC, the oxygen vacancies on the surface of cerium oxide are the catalytic active sites. O_3_ can react with oxygen vacancy directly or with H_2_O which was adsorbed on oxygen vacancy to generate HO_3_·, ·O_2_^−^, HO_2_· and other free radicals [24]. These free radicals could be transformed into ·OH through a series of chain reactions [6]. In the process of free radical generation, the Ce^3+^ around the oxygen vacancy will lose electron and be oxidized to Ce^4+^, which could result in the decrease of catalytic activity [24]. Therefore, electron transfer processes might happen between electron-rich structure (such as graphene layer, -NO_2_ and pyrone) and cerium oxide, leading to reduction of Ce^4+^ and the renewal of catalytic activity. This synergistic effect might be the key for the rapid and continuous generation of ·OH on the surface of Ce/AC.

## 3. Materials and Methods 

### 3.1. Materials 

Reverse osmosis concentrate was taken from a municipal wastewater treatment plant in which biologically treated secondary effluent was subjected to MBR+RO system to remove organics and salts for water reclamation. The characteristics of the ROC are shown in Table 4.

Activated carbon was purchased from SIGMA-ALDRICH Company with a particle size of 20–40 meshes. Tert-Butanol (TBA), p-CBA, tetracycline, metoprolol, atrazine, methanol, dichloromethane, hexane and ammonium acetate were purchased from DIKMA (Beijing, China) with the grade of chromatographic pure. The other reagents used in experimental are analytical pure. Ultrapure water was produced by Thermo SMART 2 PURE ultrapure water meter (μ = 18.2 MΩ × cm, Waltham, MA, USA).

### 3.2. Preparation of the Catalysts 

Before it was used as catalyst, AC was pretreated as the following procedures. It was firstly mixed with an acid solution (15% HCl and 5% HF) and stirred for 24 h to remove ash. After filtration, the AC was washed with ultrapure water until the pH was nearly neutral. The samples were then dried for 12 h and named AC-A to indicate “after wash of acid”. 

After pretreatment, AC was used to prepared AC-NH_2_, AC-NO_2_ and Ce/AC. 

Nitric acid modified activated carbon (AC-NO_2_) was conducted as follows. Two grams of AC-A was mixed with 40 mL CH_3_COOH in three-necked flask. Then, 30 mL of fuming nitric acid was slowly dripped into the suspension. The mixture was stirred at 0 °C for 5 h and stirred at 25 °C for 19 h. After that, the samples were washed until the filtrate was neutral and dried for 12 h. 

To prepare the aminated activated carbon (AC-NH_2_), 1g AC-NO_2_ was mixed with 30 mL aqueous ammonia (about 8%) and 0.75 g of NaBH_4_. After a 24 h agitation, the solid was filtered and washed with ultrapure water until nearly neutral. Then, the sample was dried for 12 h.

Cerium-loaded activated carbon (Ce/AC) was synthesized by impregnation method. AC-A was mixed with Ce(NO_3_)_3_ solution (AC:Ce = 5:1) and stirred for 12 h. The suspension was dried for 12 h. The dried samples were calcined in a muffle furnace at 400 °C for 4 h in N_2_ flow to obtain Ce/AC.

### 3.3. Experimental Methods 

Ozonation processes were carried out in the experimental system shown in Figure 5. The mixture of Ozone/oxygen gas was produced by an ozone generator (VMUS-ASE, AZCO INDUSTIES Ltd., Vancouver, B.C., Canada), which was then bubbled into a borosilicate glass oxidation reactor (h = 1500 mm, Φ_in_ = 40 mm) equipped with a glass porous plate at the rate of 1.5 L/min. The reaction conditions were: O_3_ dosage, 12.15 mg/min; H_2_O_2_ dosage, 142 mg/L; catalysts dosage, 0.2 g/L; volume of p-CBA solution or ROC, 500 mL; concentration of target pollutants, 2 mg/L. Water samples were collected and filtered with 0.45 μm filter membrane. Na_2_S_2_O_3_ solution was used to quench the continuous ozonation reaction in the samples.

### 3.4. Analytical Methods 

The specific surface area and porosity of all the catalysts were measured with a surface area analyzer (JSM 7401, Japan Electronics Corporation). The surface of cerium-loaded activated carbon was detected by a field emission scanning electron microscopy (JSM 6301F, Japan Electronics Corporation, Tokyo, Japan). The acid and base groups on the surface of AC-A, AC-NO_2_ and AC-NH_2_ were measured by Boehm titration method with a potentiometric titrimeter (Titration Excellence T50, METTLER TOLEDO, Zurich, Switzerland).

The concentration of p-CBA was analyzed by a HPLC (CBM-10A VP plus, Shimadzu, Kyoto, Japan) with InertSustain C18 column (5 micron, 4.6 × 250 mm, Kyoto, Japan) and UV detector (SPD-10A vp plus). The detection wavelength was 236 nm and the injection volume was 20 μL. The mobile phase was 55% methanol solution with the flow rate of 1.0 mL/min. Column temperature was control at 35 °C. TOC of samples was determined by total organic carbon analyzer (TOC-L CPH CN200, Shimadzu) with NPOC method. 

The concentration of tetracycline, metoprolol and atrazine was analyzed by a combination of reversed solid phase extraction (SPE) and HPLC-MS. Water samples were extracted with LC-C18 SPE column, and then eluted with n-Hexane and CH_2_Cl_2_ at 70:30 (*v/v*) for 3 times, 4 mL, 3 mL and 3 mL successively. In HPLC-MS system, the sampling volume was 10 μL. An InertSustain C18 column (5 μm, 4.6 × 250 mm, Kyoto, Japan) was used at 35 °C for separation of each organics. The mobile phase was a mixture of methanol and 20 mmol/L NH_4_OAc solution at 10:90 (*v/v*) and the flow rate is 0.3 mL/min. The ion source is ESI. Temperatures of DL tube and heating block are 250 °C and 400 °C respectively.

## 4. Conclusions

Four catalysts, e.g., AC-A, AC-NO_2_, AC-NH_2_ and Ce/AC, were prepared by modification and impregnation. Nitration and amination increased the concentration of surface acid and basic functional group respectively, which can improve the activity of activated carbon to some extent. Loading Ce on activated carbon can significantly enhance the removal of organic compounds from water in Ce/AC catalyzed ozonation process due to adsorption, ozonation and free radical oxidation. Ce/AC is very active in the catalytic ozonation of p-CBA in water and mineralization of organic compounds in ROC. It can greatly stimulate the generation of ·OH which plays an important role for the removal of refractory organic compounds in water. Functional groups on the surface of activated carbon is helpful for the generation of ·OH. What is more, the combination of the redox cycle of Ce^3+^/Ce^4+^ and electron transfer between electron-rich structure and cerium oxide is very efficient for the rapid and continuous generation of ·OH on the surface of Ce/AC. Because of these advantages, Ce/AC is a promising catalyst for catalytic ozonation.

## Figures and Tables

**Figure 1 molecules-24-04365-f001:**
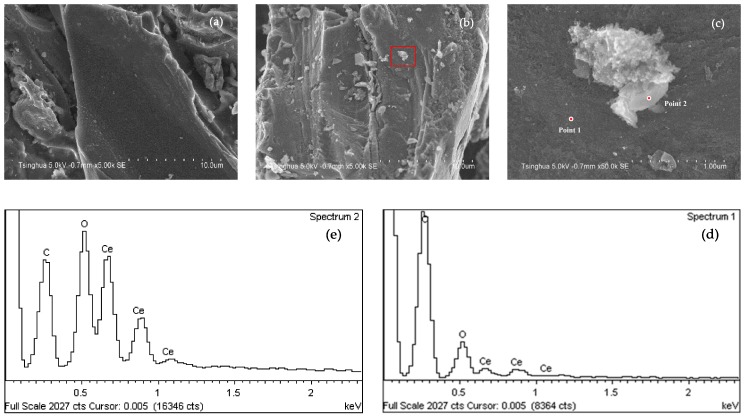
SEM and EDS analysis results: (**a**) SEM of AC-A; (**b**,**c**) SEM of Ce/AC; (**d**) EDC of Point 1; (**e**) EDS of Point 2.

**Figure 2 molecules-24-04365-f002:**
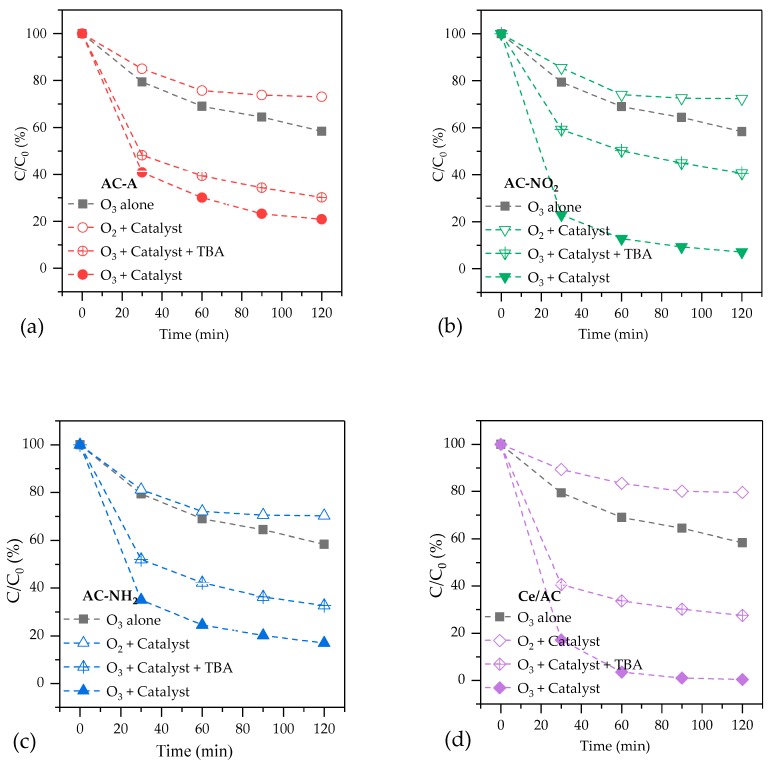
The removal of p-CBA by catalytic ozonation with four catalysts: (**a**) AC-A, (**b**) AC-NO_2_, (**c**) AC-NH_2_ and (**d**) Ce/AC.

**Figure 3 molecules-24-04365-f003:**
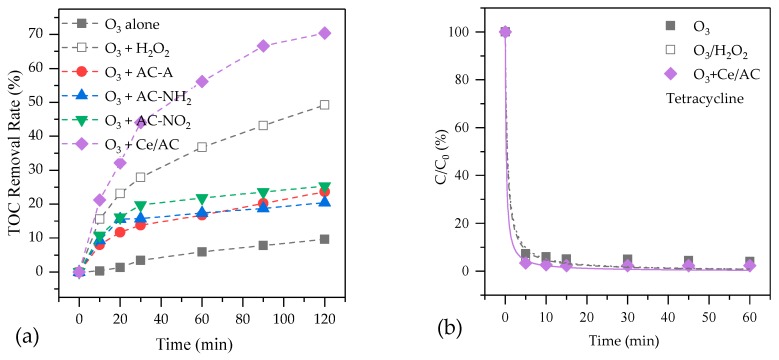
The efficiency of organic degradation by O_3_ alone, O_3_/H_2_O_2_ and catalytic ozonation processes in ROC: (**a**) TOC; (**b**) Tetracycline; (**c**) Metoprolol; (**d**) Atrazine.

**Figure 4 molecules-24-04365-f004:**
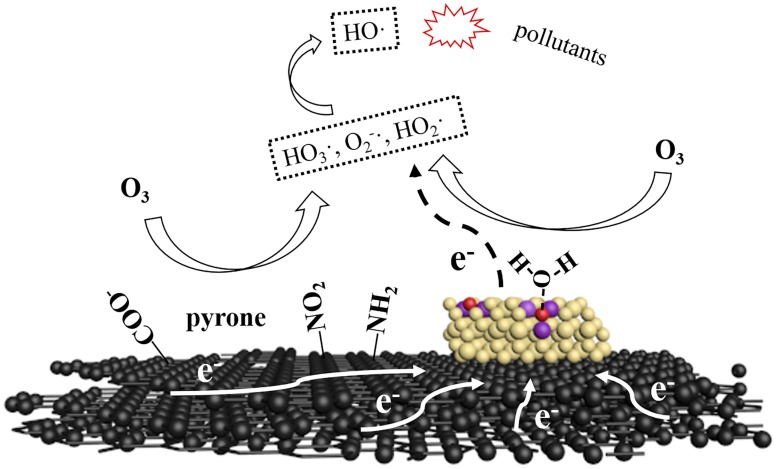
A possible mechanism for generation of ·OH in catalytic ozonation system with Ce/AC. black: graphene; brown: cerium oxide; red: oxygen vacancy; purple: Ce^3+^.

**Figure 5 molecules-24-04365-f005:**
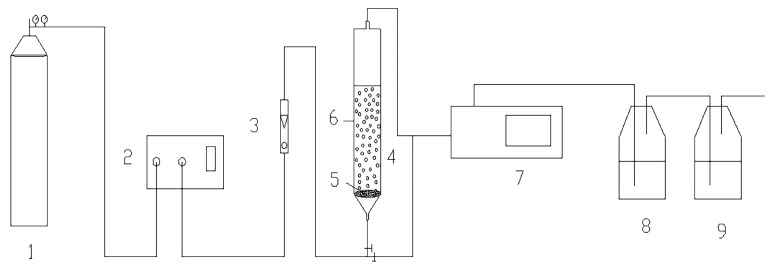
Experimental System (**1**) Oxygen bomb; (**2**) Ozone generator; (**3**) Gas flowmeter; (**4**) Tubular borosilicate glass oxidation reactor; (**5**) Aeration plate; (**6**) Sampling port; (**7**) On-line ozone concentration analyzer; (**8** & **9**) Absorption bottle.

**Table 1 molecules-24-04365-t001:** BET analysis result of different catalysts.

Catalysts	AC-A	AC-NO_2_	AC-NH_2_	Ce/AC
S_BET_ (m^2^/g)	1037.85	1030.32	1072.84	865.49
d_p_ (nm)	0.8573	0.8364	0.8368	0.8162

**Table 2 molecules-24-04365-t002:** Acid and basic surface group concentration of modified activate carbon.

Catalysts	AC-A	AC-NO_2_	AC-NH_2_
Acid Groups (μeq/g)	194.0	313.4	74.6
Base Groups (μeq/g)	153.9	57.7	283.7

**Table 3 molecules-24-04365-t003:** The reaction constants and contribution rates of ·OH reaction and surface reaction.

AOPs	Reaction Constant (×10^−3^·min^−1^)		Contribution Rate (%)	
	k	k_TBA_	·OH	Non ·OH
O_3_ + AC-A	8.20	5.28	35.6	64.4
O_3_ + AC-NO_2_	14.59	4.24	70.9	29.1
O_3_ + AC-NH_2_	8.54	5.37	37.1	62.9
O_3_ + Ce/AC	51.52	4.42	91.4	8.6

**Table 4 molecules-24-04365-t004:** Characteristics of municipal reverse osmosis concentrate (ROC).

COD (mg/L)	TOC (mg/L)	UV_254_ (cm^−1^)	pH	Conductivity (S/cm), 25 °C	TDS (mg/L)
31.44~41.86	9.28~10.69	0.19±0.05	6.5±0.5	1896	948

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
