# Peer review of "Catalytic Ozonation of Organics in Reverse Osmosis Concentrate with Catalysts Based on Activated Carbon"

_molecules, 2019, doi:10.3390/molecules24234365_

Round 1

Reviewer 1 Report

Authors studied decomposition of some organic pollutants present in water by application of catalyzed ozonation. Functionalized carbon materials have been used as catalysts and the highest activity has been noted for cerium loaded activated carbon. All catalysts were sufficiently characterized by the appropriate methods and their reactivity was evaluated on the basis of kinetic data. The reaction mechanism was proposed and the positive role of the oxygen vacancies on the surface of cerium oxide was indicated.

The paper is clearly presented. I suggest to add short information about the organic products formed by ozonation to give broader context aout the usefulness of the method.

Reviewer 2 Report

In this manuscript, modified ACs, especially Ce/AC, demonstrated a high removal efficiency of organics. The research approach is very systematic and manuscript being well written. A few comments as below:

Line 70, Give the full name of "TBA" when first introduced in the manuscript. 

Line 74, The TDS/conductivity is relatively LOW for RO concentrate. To me, it seems the secondary effluent might not need RO treatment, based on the water quality of the ROC presented.  

Line 86, check the formula "NaBH3",I believe it should be "NaBH4".

Line 155-157, interpretation of Figure 3 is not clear, suggesting paraphrase. 

Figure 3 contains data of O3+catalyst+TBA, but not explained. It should be mentioned/discussed for better understanding (not confusing readers) in section 3.2 and 3.4. 

Reviewer 3 Report

The following minor changes are suggested:

Page 3 – Line 113 – should read “injection volume were 20 rather than 20 mL.

Page 4 – Line 137 – The dp (nm) data listed in Table 2 should be explained and briefly discussed.

Page 2 – Line 76 – grammatical error.

Page 6 – Line 202 – grammatical error.

Page 8 – Line 245 – grammatical error.

The entire paper should be checked again for grammatical errors.

Reviewer 4 Report

This paper on the ozonization of reverse osmosis concentrates of waste waters by modified active carbons deals with a subject of great environmental interest. The paper is well structured, useful experimental design and deep discussion of the results. Cerium on activated carbon shows very good activity for hazardous organics removal in water. It deserves publication with minor modifications:

In Table 2, loading of Cerium on activated carbon lead to a decrease in BET surface. The authors attributed this decrease to a possible plugging of the porosity. However, it seems that it is just a dilution effect due to the cerium proportion 1:5 Ce/Ac. In fact, if the value of 1037 m2/g for Ac is mixed with a substantially low surface Cerium (1037*(1-1/6)=864) that is close to the value of Ce/AC BET surface. So, it is just a dilution effect. Line 208, there are two verbs in the same sentence “possesses shown”, it should be “shows”.
